# Peruvian Amaranth (*kiwicha*) Accumulates Higher Levels of the Unsaturated Linoleic Acid

**DOI:** 10.3390/ijms24076215

**Published:** 2023-03-25

**Authors:** Adnan Kanbar, Julia Beisel, Meylin Terrel Gutierrez, Simone Graeff-Hönninger, Peter Nick

**Affiliations:** 1Molecular Cell Biology, Joseph Kölreuter Institute for Plant Sciences, Karlsruhe Institute of Technology, 76131 Karlsruhe, Germany; 2Institute of Crop Science, University of Hohenheim, 70599 Stuttgart, Germany

**Keywords:** amaranth, functional food, *kiwicha*, ω3 fatty acids, cold stress, transcript profiling

## Abstract

Grain amaranth (*Amaranthus* spp.) is an emerging crop rich in proteins and other valuable nutrients. It was domesticated twice, in Mexico and Peru. Although global trade is dominated by Mexican species of amaranth, Peruvian amaranth (*A. caudatus*, *kiwicha*) has remained neglected, although it harbours valuable traits. In the current study, we investigate the accumulation of polyunsaturated fatty acids, comparing four genotypes of *A. caudatus* with K432, a commercial variety deriving from the Mexican species *A. hypochondriacus* under the temperate environment of Southwest Germany. We show that the *A. caudatus* genotypes flowered later (only in late autumn), such that they were taller as compared to the Mexican hybrid but yielded fewer grains. The oil of *kiwicha* showed a significantly higher content of unsaturated fatty acids, especially of linoleic acid and α-linolenic acid compared to early flowering genotype K432. To gain insight into the molecular mechanisms behind these differences, we sequenced the genomes of the *A. hypochondriacus* × *hybridus* variety K432 and the Peruvian *kiwicha* genotype 8300 and identified the homologues for genes involved in the ω3 fatty-acid pathway and concurrent oxylipin metabolism, as well as of key factors for jasmonate signalling and cold acclimation. We followed the expression of these transcripts over three stages of seed development in all five genotypes. We find that transcripts for Δ6 desaturases are elevated in *kiwicha*, whereas in the Mexican hybrid, the concurrent lipoxygenase is more active, which is followed by the activation of jasmonate biosynthesis and signalling. The early accumulation of transcripts involved in cold-stress signalling reports that the Mexican hybrid experiences cold stress already early in autumn, whereas the *kiwicha* genotypes do not display indications for cold stress, except for the very final phase, when there were already freezing temperatures. We interpret the higher content of unsaturated fatty acids in the context of the different climatic conditions shaping domestication (tropical conditions in the case of Mexican amaranth, sharp cold snaps in the case of *kiwicha*) and suggest that *kiwicha* oil has high potential as functional food which can be developed further by tailoring genetic backgrounds, agricultural practice, and processing.

## 1. Introduction

Grain amaranth (*Amaranthus*) belongs to the crop plants with the longest domestication history, especially in the New World, and was domesticated independently in Mexico and in the Andean plateau between Peru and Bolivia. The ancestral species were different. For the two Mexican amaranth species, *A. cruentus* and *A. hypochondriacus*, the wild *A. hybridus* was the progenitor, whereas the Peruvian *A. caudatus* (*kiwicha*) originated from the wild *A. quitensis* [1]. Although the Mexican amaranth served as a staple food in the culture of Aztecs and Mayans, the Peruvian *kiwicha* did not have the same impact upon the civilisation of the Inca, even prior to Spanish colonisation. *Kiwicha* was grown more sporadically, mainly for worshipping and as a robust crop to sustain subsistence agriculture. The role of the staple crop in Andean cultures was rather played by quinoa and potato [2]. As a consequence of the suppression of Inca culture by Spanish colonists, *kiwicha* became even completely forgotten until it was rediscovered and promoted by Luis Sumar Kalinowski (1993). This may be the reason why *kiwicha*, compared to its Mexican relatives, was not as much altered by breeding and in some respect even represents a state of incomplete domestication [1]. Since just over a decade ago, amaranth has become popular in the West due to its outstanding nutritional value, fuelling a rapidly growing demand in industrialised countries [3]. For Europe alone, the annual market volume for amaranth was estimated to exceed 2000 million EUR already in 2020 and is projected to reach 5000 million EUR by 2028 with the growing vegan population in this continent [4].

Grain amaranth in general [5] and *kiwicha* in particular [6] have great potential as a functional food. For instance, amaranth seeds contain more lysine than other staple crops, rendering them valuable as a vegan alternative to animal protein. In addition, it is rich in provitamin A, as well as oil, reaching up to 6–9% [7]. The oil contains high levels of palmitic (>20%), oleic (>25%), and linoleic acids (>40%), as well as minor amounts of the polyunsaturated fatty acids α-linolenic (>3%), arachidic (>5%), and trace amounts of gadoleic and behenic acids [8]. Moreover, the amount of α-linolenic acid in amaranth is comparable to cereals such as wheat (3–5%) and barley (4–6%) [9]. Moreover, several studies even reported the presence of docosahexaenoic acid (C22:6) in amaranth oil (5.6–21.4%; [10] 6.7–7.4%; [11]), and small amounts of eicosapentaenoic acid (1.8–2.8%) and docosahexaenoic acid (1.0–1.4%) were even found in amaranth flowers [12]. However, other studies failed to detect docosahexaenoic acid in amaranth seed oil [13,14,15]. The reason for these discrepancies has remained unclear because the taxonomic identity of the source material has been rarely assessed in those studies.

Especially docosahexaenoic acid (DHA) would be of great interest, since this essential fatty acid has been found to be crucial for neural development, especially in the foetal brain [16], and since it is rarely found in plants. In addition, ω3 fatty acid intake is recommended to prevent cardiovascular events, particularly in patients with multiple cardiovascular risk factors [17] and also can reduce depressive symptoms and exert anti-inflammatory action [18].

The content of polyunsaturated fatty acids depends on the regulated activity of fatty acid desaturases (FADs) and is relevant to maintaining the fluidity of plant membranes under low temperatures [19]. Their biosynthesis has been intensively studied in plants [20] and initiates in the plastid, where coenzyme A-activated acetate residues are assembled to stearic acid (18:0), which is then conjugated to desaturated by a stearoyl-ACP desaturase (SAD) generating oleic acid (18:1). The crucial step is the transition from oleic acid with one double bond (18:1) to linoleic acid (18:2) by FAD2 (or Δ12 fatty acid desaturase), introducing a second double bond and, thus, initiating the biosynthesis of polyunsaturated lipid synthesis in plant tissues [21,22]. In fact, FAD2 transcripts are induced by low temperatures in a number of plants [23]. Linoleic acid (18:3) is then converted in the next step by FAD7 (or Δ15-desaturase) to form the polyunsaturated α-linolenic as the first committed step of the ω3 fatty acids [24]. Additionally, for this enzyme, gene expression has been reported to increase in response to cold stress, for instance in olive [25], or in *Arabidopsis thaliana* [26]. From here, alternating reactions of desaturation and elongation give rise to the other ω3 fatty acids. For instance, FADΔ6 (Δ6 fatty acid desaturase) converts α-linolenic acid to stearidonic acid. This desaturase was first identified in *microalgae* but is also present in higher plants [27]. The resulting stearidonic acid is then elongated to Eicosatetraenoic fatty acid by Δ6 fatty acid elongase [24]. Then, the desaturation in position 5 gives rise to eicosapentaenoic acid (EPA), which is then elongated into docosapentaenoic acid (DPA) by a Δ5 elongase. Eventually, DPA is desaturated in position four to yield DHA by means of a Δ4-desaturase [28,29]. Whether homologues for these three enzymes of the terminal ω3 biosynthetic pathway (Δ5 desaturase, Δ5 elongase, and Δ4 desaturase) exist in amaranth is not known, however.

However, this pathway towards polyunsaturated fatty acids has a concurrent branch diverging from α-linolenic acid, which can be oxidised by plastidic lipoxygenases (LOXs) giving rise to oxylipins, including jasmonates (JA), central stress signals of land plants [30]. The product, 13-hydroperoxy octadecatrienoic acid is either processed by allene oxide synthase further towards jasmonates or turned down by the closely related hydroperoxylyase into volatile aldehydes, such as cis-3-hexenal that can induce programmed cell death [31]. Additionally, jasmonate synthesis is responsive to cold [32], and jasmonate signalling was found to interfere with cold signalling, namely, the inducer of CBF expression 1 (ICE1). This master regulator for cold-stress signalling activates a transcriptional cascade through inducing the transcription of the cold box factors (CBFs) that, in turn, regulate cold-responsive genes (CORs) through the promoters of their target genes [33]. Specific jasmonate response factors (so-called JAZ genes) form a complex with ICE1, such that it prevents binding to the promoters of its targets, the CBF genes, which disrupts cold-dependent gene expression [34].

Thus, under cold stress, α-linolenic acid can undergo three fates: (1) It can fuel the synthesis of polyunsaturated acids, which helps to sustain membrane fluidity under low temperatures and must be considered as an adaptive phenomenon. (2) It can be oxidised by lipoxygenase in response to reactive oxygen species resulting from perturbed photosynthesis, being metabolised into jasmonates. This will interact with the expression of cold-responsive genes. This is of an adaptive nature as well, but far more demanding to the plant since it requires the extensive re-programming of gene expression. (3) It can be converted by lipoxygenase and hydroperoxylyase into volatile compounds that activate the programmed cell death of the cell, which allows the plant to mobilise and repartition resources for other parts of the plant. This self-sacrifice of the stressed organ is certainly the most expensive strategy to respond to cold stress and must be considered as a kind of last resort.

Attempts to cultivate amaranth in temperate climates, such as in Europe, are often limited by the fact that grain amaranths mature late in the season, and, thus, seed filling falls into late autumn, such that the plants face cold stress [35,36]. The ability to safeguard unsaturated fatty acids becomes relevant here because membranes rich in lipids with polyunsaturated fatty acids can sustain fluidity even when the temperature drops [37]. Membrane fluidity is a central precondition for the functionality of electron transport in both photosynthesis and respiratory metabolism [38,39]. In fact, there are many examples, where the content of unsaturated fatty acids in lipid membranes increases with decreasing temperature [19,40,41,42,43,44]. Our hypothesis is that Peruvian amaranth *kiwicha* (*A. caudatus*), due to its domestication in high altitudes, is endowed with a higher cold tolerance linked with the accumulation of higher levels of unsaturated fatty acids and a lower level of oxylipins. To address this hypothesis, we have prioritised a set of four genotypes of *A. caudatus* in comparison to the Mexican species *A. hypochondriacus* from an extensive screen for phenotypic traits such as plant height, flowering time, or inflorescence length [45]. We have raised these five accessions side by side under the temperate conditions of southwest Germany following the expression of selected genes involved in the synthesis of polyunsaturated fatty acids, along with key markers for the concurrent oxylipin synthesis and markers for the cold-stress response over three stages of seed development. In parallel, we have analysed oil contents and fatty acid profiles for the same genotypes under the same conditions. We can show that *kiwicha*, in contrast to Mexican amaranth (*A. hypochondriacus*), can channel fatty acid biosynthesis towards polyunsaturation, evading lipoxygenation and oxylipin synthesis. At the same time, *kiwicha* can deploy cold adaptation, as evident from the accumulation of transcripts for Cold Box Factor 4 and the dehydrin Xero2. Our findings highlight that genetic identity is relevant for the value-giving compounds of amaranth oil and that insight into the functional aspects of these value-giving compounds is relevant to tailor amaranth cultivation for the use of the product as a functional food.

## 2. Results and Discussion

Amaranth was domesticated twice under different climatic conditions and from different ancestral species, stimulating the question of whether these differences are reflected in the fate of polyunsaturated fatty acids (PUFAs). In plants, the PUFA α-linolenic acid can feed two concurrent pathways—one leading to highly desaturated fatty acids, such as DHA, and the other through peroxidation, resulting in the accumulation of oxylipins, including the important stress signal jasmonate. The two centres of domestication differ strongly with respect to temperature leading to the question of whether the partitioning of PUFAs between the concurrent fates might differ. Therefore, we analysed the expression of key genes in the ω3 and the jasmonate pathway, as well as of markers for the response to cold stress during three defined developmental stages of seed development and integrated these findings into measurements of oil content, fatty acid profiles, and morpho-physiological parameters.

### 2.1. Kiwicha Is Taller and Flowers Later under the Temperate Conditions of Germany

Since the prioritised set of five amaranth genotypes (four *A. caudatus* from Peru and one *A. hypochondriacus* × *hybridus* from Mexico) was expected to respond differently to the temperate conditions in Southern Germany with long days, we started off by determining the agronomically relevant phenotypic traits, including the quantitative traits time to flowering, plant height at maturity, inflorescence length, grain yield per plant, 1000-seed weight, and cross area of seed (Figure 1), and the two qualitative traits seed colour and flower colour (Figure 2).

The mean values of agro-morphological traits recorded from the tested genotypes, together with their grouped overall mean values, least significant differences, and coefficient of variation within the trait are presented in Appendix A. ANOVA showed that all the tested quantitative traits varied significantly (*p* < 0.0001) between the screened genotypes (Appendix A). Our experiment showed that the four *A. caudatus* genotypes differed significantly from the Mexican genotype by delayed flowering, taller stature, longer inflorescence, and higher 1000-seed weight (Figure 1A–C,E). However, the grain yield per plant was significantly lower for all *A. caudatus* genotypes as compared to the *A. hypochondriacus* × *hybridus* genotype K 432 (8039) (Figure 1D). The time required for flowering is most crucial because it decides to what extent seed development is confronted with the dropping temperatures in autumn. Here, the *A. hypochondriacus* × *hybridus* genotype K 432 (8039) required only 79 days to reach 50% of flowering, whereas the *A. caudatus* genotype 2496 started flowering only 10 days later, and the economically relevant *A. caudatus* genotype Oscar Blanco 2 (9445) flowered even only after 98 days, which is 20 days later than K 432 (8039).

In general, the *Amaranthaceae* are considered to be indeterminate with respect to day length [46]. However, *A. caudatus* exhibits distinct photoperiodism [47]. Once it has reached its sensitive period, which occurs around 30 days after germination, 2 short days (9 h light) can induce floral primordia, whereas under long days (18 h light), it requires 60 days to initiate flowering. It should be noted that these experiments [47] were carried out during an Illinois summer with night temperatures above 25 °C, so the development was much faster than in the current experiment, where the temperature was much lower (Appendix A). Our observations are consistent with this. The fact that *A. hypochondriacus* lacks evident photoperiodism is not surprising, since it originates from a region where daylength varies maximally by a bit more than an hour. More surprising is the fact that *A. caudatus* produces such a pronounced photoperiodism. The daylength in Peru is fairly equal. However, its ancestral species *A. quitensis* [48] extends to regions that are far more distant from the equator (www.gbif.org/species/7323711, accessed 15 November 2022) and where daylengths are distinct depending on the season. It is straightforward to assume that *kiwicha* is a (non-obligate) short-day plant because its ancestor was. Under the even daylength of Peru, this is not of further relevance, but under the distinct seasonality in Southern Germany, it is.

The differences in flowering time were also reflected as differences in plant height. The lowest plant height (109 cm) was recorded in the *A. hypochondriacus* × *hybridus* genotype K 432 (8039), whereas all *A. caudatus* genotypes were approximately twice as tall (Figure 1B). The pattern for inflorescence length paralleled that seen for total plant height only partially. First, all five tested genotypes were significantly different among themselves (*p* < 0.0001), as shown in Appendix A. Again, for the *A. hypochondriacus* × *hybridus* genotype K 432 (8039), inflorescence length was the lowest (60 cm), whereas for *A. caudatus* genotypes Peter 1 (8300), Peter 2 (8301), inflorescences were above a meter in length (Figure 1C). Instead, for the two other accessions for *A. caudatus*, Marron (2496) was clearly shorter, and for Oscar Blanco 2 (9445), there was no significant difference with *A. hypochondriacus* × *hybridus* genotype K 432 (8039). Again, a link with photoperiodism is to be expected. Among the three cultivated grain amaranths (*A. caudatus*, *A. cruentus* and *A. hypochondriacus*), only *A. caudatus* is photosensitive [49,50,51], whereas *A. cruentus* and *A. hypochondriacus* are insensitive amaranths [52]. Under the long-day conditions of Central Europe, short-day crops such as *A. caudatus* delay flowering, prolong biomass accumulation, and increase plant height [53,54]. This is consistent with our observations, but one needs to consider that the relation between total height and the length of the inflorescence is not necessarily coupled but depends on the relative timing between vegetative shoot elongation and the onset of flowering. It might be that the different *A. caudatus* genotypes differ with respect to internode elongation during the vegetative phase.

Additionally, for grain yield per plant, the five genotypes evaluated were significantly (*p* < 0.0001) different from each other (Appendix A). Here, the *A. hypochondriacus* genotype K 432 (8039) performed better (17.3 g/plant) under the temperate environmental conditions than all four *A. caudatus* genotypes (Figure 1D). Especially for the two genotypes with shorter inflorescences, Oscar Blanco 2 (9445), with only 2.7 g/plant, and Marron (2496), with only 2.1 g/plant, this value was very low, whereas the genotypes Peter 1 (8300) with 10.7 g/plant and Peter 2 (8301) with 7.8 g/plant accumulated more. Under long photoperiods of the temperate climate, as it prevails in Southwest Germany, grain amaranth is limited by late flowering, late grain maturation, and slow plant dry-down leading to harvest difficulties and grain losses [35,36]. Therefore, early flowering would be more advantageous to minimise exposure to low temperatures on the sensitive flowering and post-anthesis grain-filling periods. Compared to studies conducted in East Austria [33,55,56,57], *A. caudatus* yielded lower in our study, which is certainly due to the more temperate conditions at our study site, where night temperatures dropped below 6 °C and 4 °C already in the months of September and October (Appendix A). Likewise, studies from Africa and Southern Italy [57] concur with the strong impact of temperature. This is easy to understand, since in different amaranth species, net photosynthesis increases almost linearly between 10 °C and 30 °C and decreases when the temperature is lower, typical behaviour for a C4 crop [58,59,60,61,62].

The strong differences in grain yield are not reflected in the size of the seeds. Both 1000-seed weight (Figure 1E) and the cross area of seed (Figure 1F) did not correlate with the grain yield per plant and differed around 25–30% at most between the genotypes. For instance, *A. caudatus* ‘Oscar Blanco 2′ (9445), which was very low-yielding, had the largest value for 1000-seed weight and also for cross area, whereas *A. hypochondriacus* × *hybridus* K 432 (8039), which showed by far the highest yield per plant, showed the lowest value for 1000-seed weight and for cross area. The values found in our study fell into the ranges reported previously for more continental conditions of Austria [55,63] and Slovakia [55,63,64]. In an extensive comparative study with 84 amaranth genotypes belonging to different species, conducted under the same conditions [45], we found that around a third of these accessions had 1000-seed weights higher than those observed here, mostly genotypes belonging to the Mexican species *A. cruentus* and *A. hypochondriacus.*

We also studied two qualitative traits, seed colour and flower colour (Figure 2). Whereas *A. caudatus* ‘Marron’ (2496) showed seeds of black colour, the other three *A. caudatus* genotypes had white seeds, and *A. hypochondriacus* × *hybridus* K 432 (8039) had cream-coloured seeds. The flowers of all four *A. caudatus* accessions were of a light pink hue, whereas the flower of *A. hypochondriacus* × *hybridus* K 432 (8039) was purely white. Seed colour is considered as important domestication trait in grain amaranths, and many breeding programs try to develop high-yielding amaranth varieties with light seed colour [65]. Based on the darker colouration of many accessions, *A. caudatus* is often considered as less advanced in domestication [48]. The black seed colour for *A. caudatus* ‘Marron’ (2496) might indicate that seed colour was not in the scope of selection during domestication. However, this conclusion must be drawn with care. Although white seeds are predominant in cultivated amaranth, which most likely represents a selected trait, other seed colours may have been preferred for different uses [48].

### 2.2. Kiwicha Shifts the Saturated Palmitic Acid into the Unsaturated Linoleic Acid

The study was motivated by the hypothesis that, in contrast to Mexican amaranth, the domestication of *kiwicha* (*A. caudatus*) took place under conditions where cold stress, especially during the crisp Andean nights, was a shaping factor. As membrane fluidity in plants under cold stress is sustained by increasing the content of phospholipids harbouring unsaturated fatty acids, a testable implication of our hypothesis would be a higher content of unsaturated fatty acids in amaranth oil. In fact, we found a significant (*p* < 0.01) difference in the measured fatty acids between the screened genotypes (Appendix A), although variation between the replications of a given genotype was insignificant. The oil content in the five genotypes ranged between 5.97 and 7.13%, with the highest value found in *A. caudatus* Oscar Blanco 2 (7.31%), followed by *A. hypochondriacus* × *hybridus* K432 (6.61%), *A. caudatus* Peter 1 (6.52%), and *A. caudatus* Peter 2 (6.37%), and the lowest value in *A. caudatus* Marron (5.97%). These values are in line with the 5.7–9.0% reported by the literature for seed oil in grain amaranths [66,67,68,69,70] and contrast with the oil content reported for wild species from Nigeria with values between 9.75% in *A. arthropurpureus*) and 16.95% in the weed *A. spinosus* [71]. Instead, our values are consistent with a large-scale screening of 104 genotypes from 30 amaranths which found oil contents between 1.9% and 8.7% [8]. In some oil crops, such as rapeseed and flax, oil content can increase with decreasing temperature during seed maturation, whereas this phenomenon is not seen in other oil crops, such as sunflower, safflower, and castor bean [72]. If this phenomenon existed in grain amaranth, we would expect a higher oil content in *kiwicha*, where seeds developed significantly later than in the early-flowering *A. hypochondriacus* × *hybridus*. This is obviously not the case, indicating that oil content does not depend on temperature; this does not exclude the possibility that oil fatty acid profile does.

In fact, we were able to detect and quantify some saturated fatty acids (Appendix A) including palmitic acid myristic acid (C14:0), palmitic acid (C16:0), stearic acid (C18:0), arachidic acid (C20:0), behenic acid (C22:0), and lignoceric acid (C24:0). For all of these saturated fatty acids, the content was significantly higher in the *A. hypochondriacus* × *hybridus* genotype as compared to the four *kiwicha* genotypes. With the exception of myristic acid, which was found only in small amounts (0.18–0.24%), the abundance of the respective saturated fatty acid decreased with the chain length. Whereas palmitic acid (C16:0) accounted for >20% of the oil, the abundance of lignoceric acid (C24:0) was two orders of magnitude lower. However, irrespective of the overall abundance, all *kiwicha* genotypes accumulated less of the respective saturated fatty acid. Our values are congruent with data from the literature [73,74,75]. For instance, palmitic acid as a preponderant saturated fatty acid in amaranth oil was also reported by other studies ([76] 19.1–23.4%; [14] 21.4–23.8%).

When we searched for unsaturated acids, we were not able to detect either eicosatetraenoic acid (C20:4), eicosapentaenoic acid (C20:5), docosapentaenoic acid (C22:5), or docosahexaenoic acid (C22:6), even if we repeated the analysis using a GC column with opposite polarity. We also found only small amounts of palmitoleic acid (C16:1), which differed between the genotypes, whereby the three *kiwicha* accessions, 8300 (Peter 1), 8301 (Peter 2), and 2496 (Marron), had only half of the levels seen in *A. hypochondriacus* × *hybridus* (8039) and the *kiwicha* variety Oscar Blanco 2 (2496). However, we found significant amounts of C18 unsaturated acids (Figure 3). Hereby, linoleic acid (18:2) dominated, amounting to around half of the oil (Figure 3B), whereas its precursor, oleic acid (18:1) accounted for around 20% of the oil (Figure 3A); also relevant (~0.8%) was the three-fold unsaturated α-linolenic acid (18:3) (Figure 3C), whereas stearidonic acid (18:4) was below 0.1% (Figure 3D). The most salient difference was a significantly increased level of linoleic acid in all *kiwicha* accessions compared to *A. hypochondriacus* × *hybridus*, which harboured around 3% less of this polyunsaturated fatty acid (Figure 3B). If one compares the levels of the four measured unsaturated C18 fatty acids (Figure 3) with their saturated precursor stearic acid (Appendix A), significant differences can be discerned: in *A. hypochondriacus* × *hybridus*, C18:0 is the lowest of all values, whereas C18:1 is the highest, and C18:2 is the lowest again. This indicates that the SAD introducing the first double bond is active, whereas the FAD2, responsible for the subsequent desaturation step, is not. For the three *kiwicha* accessions, 8300 (Peter 1), 8301 (Peter 2), and 2496 (Marron), the pattern is inversed—here, the reaction catalysed by FAD2 is so active that the steady-state levels of the precursor oleic acid are depleted, although the pool of the precursor C18:0 is high. The genotype Oscar Blanco 2 falls out of this pattern because, here, the activity of FAD2 inserting the second double bond is much lower than in the other three *kiwicha* genotypes and almost matches the pattern seen in *A. hypochondriacus* × *hybridus*. Thus, although the majority of *kiwicha* accumulate higher levels of linoleic acid than *A. hypochondriacus* × *hybridus*, indicative of a higher FAD2 activity, there are significant differences within *kiwicha*, which means that the choice of genotype is a significant factor for the composition of the oil.

The linoleic acid contents of grain amaranth species other than *A. caudatus* had been tested in a comparative study by [76]. Here, only *A. hypochondriacus* and *A. tricolor* were found to accumulate contents of close to 50%, the other species, *A. cruentus* and *A. hybridus* showed values that were 5–10% lower. In other words, *kiwicha* excels all grain amaranths with respect to linoleic acid content. Likewise, a comparative study on Mexican grain amaranths under Central European conditions [77] found linoleic acid levels that were the lowest for *A. cruentus* and somewhat higher for *A. hypochondriacus*, albeit not reaching the levels seen for *kiwicha* in the current study. The data obtained for *A. hypochondriacus* × *hybridus* K 432 in the study by [14] for linoleic acid almost exactly matched the values found for this genotype in our study, indicative of a strong genetic component.

However, there seems to be more impact of the environment with respect to α-linolenic acid. Here, the levels obtained for *A. hypochondriacus* (*A. caudatus* was not included in that study) under the conditions of a Central European summer [77] were around 0.8% and, thus, were quite close to the values found in the current study, whereas in the study by [76], conducted under the conditions of a warmer and continental Chinese summer, the total amount of polyunsaturated fatty acids remained below 0.3%. The α-linolenic acid levels in *A. hypochondriacus* × *hybridus* K 432 under the continental conditions of Idaho [14] were significantly higher (0.9%). This points to an influence of environmental conditions—under the temperate conditions of Europe ([77]; the current study), α-linolenic acid seems to accumulate at higher levels as compared to the situation of a Chinese summer, and in Idaho, where night temperatures in late summer drop much more rapidly than in Europe, α-linolenic acid seems to be even higher. Whether there is a link between α-linolenic acid levels and night temperatures at seed maturation remains to be elucidated. Classical studies have shown that oil crops respond differently to lower temperatures [72]. Although castor bean and safflower, both domesticated in subtropical to tropical regions, do not seem to be responsive, crops that are adapted to cooler climates, such as flax or rapeseed, increase the abundance of unsaturated fatty acids in response to lower temperatures. This response allows for maintaining membrane fluidity when temperature drops, since unsaturated fatty acids, due to their kinked structure, occupy more space and counteract rigidification. As the *A. hypochondriacus* × *hybridus* originated in Mexico, it is not surprising that it accumulates less linoleic acid as compared to *kiwicha*, which originates from the harsher climate of the Andes. The genetic potential of *kiwicha* to accumulate linoleic acid and, possibly, fatty acids with even higher degrees of desaturation might be unfolded further to grow this crop under specific climatic conditions. We have, therefore, launched experiments, where the same genotypes are raised under different climate regimes in parallel.

### 2.3. The Genome of A. caudatus Exhibits a Higher Level of Heterozygosity

As the content of fatty acids in *Amaranthus* was clearly dependent on genotype as a manifestation of differences in the response to low temperature, we ventured to analyse the expression of genes in the metabolism of unsaturated acids, but also the concurrent oxylipin pathway and response, as well as key genes involved in the response to cold stress. To mine for these genes, we decided to sequence the genomes from one *kiwicha* accession *A. caudatus* Peter 1 (8300) along with the *A. hypochondriacus* × *hybridus* variety K 432 (8039) by 150-bp paired-end Illumina whole-genome sequencing on a NovaSeq platform mapping the genome data on the *A. hypochondriacus* PI 558,499 reference genome [78]. We were able to obtain 3,703,021 for *A. caudatus* and 1,592,370 for *A. hypochondriacus* × *hybridus* (Appendix A). As expected, the number of SNPs, insertions, and deletions with respect to the reference genome from *A. hypochondriacus* was higher for the *A. caudatus* genotype, as compared to *A. hypochondriacus* × *hybridus*. We identified an average of 3,071,634 SNPs, 270,883 insertions, and 353,977 deletions in *A. caudatus* 8300, but only 1,243,281 SNPs, 147,099 insertions, and 199,155 deletions in *A. hypochondriacus* × *hybridus* K 432 (Appendix A).

A previous study comparing the genomes of seven grain amaranths (five *A. hypochondriacus*, one *A. cruentus*, and one *A. caudatus* from Bolivia) and *A. hybridus* as their putative ancestor found 7,184,636 SNPs for the comparison over the four species, whereas the number within *A. hypochondriacus* was much lower with 1,760,433 SNPs [79]. Thus, the number of SNPs seen for *A. hypochondriacus* × *hybridus* K 432 as compared to the *A. hypochondriacus* reference genome is well congruent with those findings. Likewise, the fact that the *A. caudatus* 8300 genome comprises around half of the genome divergence from *A. hypochondriacus* found for the entire set of the four species [79] shows that our results match well with the published record.

Heterozygosity was higher in *A. hypochondriacus* × *hybridus* K 432 (836,128 heterozygous variants) as compared to *A. caudatus* 8300 (517,057 heterozygous variants), which is not surprising since K 432 derives from interspecific hybridisation between *A. hypochondriacus* and *A. hybridus*. The weedy species *A. hybridus* is significantly more heterozygous as compared to domesticated species of amaranth [79]. The lower heterozygosity in the domesticated species may derive from a lower outcrossing rate that is estimated to be around 10% [79,80]. In fact, the homozygous variants in *A. caudatus* 8300 dominated (3,185,964, almost four-fold as compared to K 432). Based on morphological and molecular data [45], this genotype has been shown to be truly *A. caudatus*, clearly differentiated from the other grain species, *A. cruentus* and *A. hypochondriacus*. The high homozygosity indicates a long history of inbreeding, either as a manifestation of geographic isolation as a landrace, or as a reflection of the intentional breeding of an improved variety.

Based on homology searches, we could predict and annotate a total of 23,883 genes for the genome of the two varieties (Appendix A). Based on the reference sequence, we generated pseudogenes using GATK v4.1.2. The files for the two genotypes containing all genes will be made available upon request.

### 2.4. All Genotypes of A. caudatus Express Higher Levels FAD2

Based on the genomes, we were able to infer the sequences for five genes of the ω3 pathway (FAD2, FAD7, FADΔ6, PSE1, and PAS2), four genes from jasmonic acid synthesis (LOX, AOS, AOC) and signalling (JAZ1), and four genes linked with the signalling of (ICE1, CBF4) and adaptation to (Xero2, RCI2) cold stress. This allowed us to follow the expression of these genes during the three stages of seed development (S1 directly post-anthesis, S2 milky, S3 full maturity) measuring the transcript abundance by RT-qPCR for the five amaranth genotypes raised under temperate conditions in southwestern Germany in 2020.

This time course study showed that, compared to *A. hypochondriacus* × *hybridus* K432 (8039), all four *kiwicha* genotypes expressed higher levels of transcripts for FAD2, encoding the D12 desaturase converting oleic acid to linoleic acid, the first polyunsaturated fatty acid of the pathway (Figure 4). For the *A. caudatus* genotype Peter 1 (8300), this difference was noted already from the beginning of seed development, whereas the *A. caudatus* Marron (2496) genotype showed a later and transient induction of FAD2 transcripts at the milky stage, and the other two *A. caudatus* genotypes induced these transcripts only in the last stage, but then to a larger extent (around five times higher than in *A. hypochondriacus* × *hybridus*, where no induction of FAD2 was observed). Thus, there is a differential expression of FAD2 between the Peruvian and the Mexican species of amaranth, and the transcript patterns are consistent with the consistently elevated levels of linoleic acid found in mature seeds of all four *kiwicha* genotypes.

For the subsequent steps of the ω3 fatty-acid pathway, such a correlation was missing: For instance, the Δ15 desaturase (FAD7) converting linoleic acid to α-linolenic acid was generally well expressed in *A. caudatus* 2496 (Figure 4B), although this genotype showed the lowest level of α-linolenic acid of all the tested genotypes (Figure 3C). Although *A. hypochondriacus* × *hybridus* K432 (8039) and the *A. caudatus* genotypes Peter 1 (8300) and Oscar Blanco 2 (9445) accumulated the same level of α-linolenic acid (Figure 3C), the two *kiwicha* genotypes accumulated substantially higher levels of FAD7 transcripts during stage 1, whereas *A. hypochondriacus* × *hybridus* K432 (8039) sustained higher levels at the later stages (Figure 4B).

The Δ6 desaturase (FAD6), converting α-linolenic acid into stearidonic acid, was highly expressed in the *kiwicha* genotypes Peter 1 (8300), Peter 2 (8301), and Oscar Blanco 2 (9445), and to a lower extent also in genotype Marron (2496). In contrast, the transcript level was significantly lower in *A. hypochondriacus* × *hybridus* K432 (8039) (Figure 4C). However, this expression was not accompanied by corresponding differences in the expected product, stearidonic acid (Figure 3D), which was low in all five genotypes. Thus, a high steady-state level of a transcript does not necessarily mean that the respective enzyme is expressed (or active) to generate the predicted product. The D6 elongase (PSE1), prolonging stearidonic acid into eicosatetraenoic acid (EHA) was detected only in low levels (Figure 4D), which would be expected by the failure to find significant levels of higher-order polyunsaturated fatty acids.

Thus, there seems to be a bottleneck at the transition from α-linoleic acid into stearidonic acid, although the transcripts of the respective enzyme (FAD6) are present and accumulate to the highest level of all transcripts tested for this pathway. Whether this bottleneck is due to posttranslational regulation or substrate sequestration remains to be elucidated.

When interpreting those data, one needs to keep in mind that, due to the differences in flowering time, the temperatures during seed maturation differed (Appendix A). Although the milky stage of *A. hypochondriacus* × *hybridus* K432 (8039) occurred at a minimal night temperature of 13.7 °C in August, the *A. caudatus* genotypes Peter 1 (8300), Peter 2 (8301), and Oscar Blanco 2 (9445) reached this stage only in September, facing a minimal night temperature of only 5.9 °C. As polyunsaturated fatty acids are needed to maintain membrane fluidity under low temperatures [19], the stronger expression of FAD2 in some of the *kiwicha* genotypes, Peter 1 (8300), Peter 2 (8301), and Oscar Blanco 2 (9445), especially during the terminal stage S3, might be understood as a response to the sinking night temperatures. This hypothesis is challenged by the fact that in the *A. caudatus* genotype Marron (2496), which developed earlier, the maximal expression of FAD2 occurred at stage 2, in the beginning of September, when it was still warm. Moreover, the *A. caudatus* genotype Peter 1 (8300) showed the highest expression at anthesis, which was in the beginning of August, when night temperatures were maximal. Thus, lower night temperatures as a consequence of later flowering cannot account for the generally stronger expression of FAD2 expression and, hence, the higher accumulation of linoleic acid in *kiwicha*. There is a strong genetic component driving this expression. On the other hand, a role for low temperature cannot be ruled out, which is also consistent with the published record. In olive trees, for instance, FAD2 and FAD7 were expressed more strongly under cold [25]. The same holds true for FAD2 in cotton [40].

The expression of very-long-chain (3R)-hydroxyacyl-CoA dehydratase PASTICCINO 2 gene (PAS2) was also measured in our experiment (Appendix A). This gene can introduce a double bond into very long fatty acids [81]. Transcripts for PAS2 were strongly upregulated in the *kiwicha* genotypes Peter 1 (8300), Peter 2 (8301), and Oscar Blanco 2 (9445), especially during the first two stages of seed development. Whereas the functional consequences of the enhanced expression of PAS2 remain unknown, this upregulation cannot be explained in terms of lower night temperatures for the same reasons as discussed above for FAD2. Thus, the stronger induction of PAS2 is also mainly under genetic rather than environmental control.

The strong genetic component in the activation of FAD2 and, hence, the accumulation of linolenic acid might be linked with the fact that *A. caudatus* can cope with the high altitudes in its native region, Peru, where it is grown at altitudes between 1500 and 3600 m above sea level [82], at temperatures that can become far lower than in Mexico, where *A. hypochondriacus* × *hybridus* K432 (8039) originates. Whether the strong induction of PAS2 in *A. caudatus* has a similar functional context is not known as long as the target remains elusive. Irrespective of this aspect, *A. caudatus* is endowed with cold tolerance and is known to tolerate chilling down to 4 °C [83]. This motivated us to probe for the response of cold stress-related markers.

### 2.5. Kiwicha Is Endowed with a More Favourable Metabolic Response to Cold Stress

Polyunsaturated acids are key factors for cold tolerance because they sustain the fluidity of membranes [19,84]. However, the double bonds are prone to oxidation, for instance by plastidic lipoxygenases, giving rise to oxylipins, including jasmonates, central signals in the adaptation to cold stress. In fact, we observed that the expression of this lipoxygenase gene was higher for *A. hypochondriacus* × *hybridus* K432 (8039) than in the *A. caudatus* genotypes, irrespective of the stage of seed development (Figure 5A). Only in stage 3, this gene was induced in *A. caudatus* as well (with the exception of genotype Peter 1 (8300), where expression remained low). This late induction of lipoxygenase transcripts in *A. caudatus* coincided with lower temperatures as compared to the early flowering *A. hypochondriacus* × *hybridus* K432 (8039) (Appendix A). This early activation of lipoxygenase transcripts in *A. hypochondriacus* × *hybridus* at still fairly warm temperatures along with the decreased steady-state levels of linoleic acid (Figure 4B) supports a scenario where polyunsaturated acids are recruited for oxylipin biosynthesis in response to low temperature, whereas *kiwicha* even being exposed to harsher temperatures due to later flowering can continue to accumulate linoleic acid rather than feeding oxylipin biosynthesis. The early induction of lipoxygenase in *A. hypochondriacus* × *hybridus* is accompanied by the accumulation of transcripts for allene oxide synthase (AOS), the first committed step of jasmonate biosynthesis (Figure 5A). Again, the response of the *A. caudatus* genotypes initiates at later stages of development (from stage 2) while reaching higher final amplitudes in genotypes Peter 1 (8300) and Oscar Blanco 2 (9445). Compared to lipoxygenase and AOS, the pattern for allene oxide cyclase (AOC) does not contrast as clearly between *A. hypochondriacus* × *hybridus* and *A. caudatus* (Figure 5C). This is consistent with the published record, where the prolonged activation of the jasmonate pathway in response to abiotic [85] and biotic [86] stress factors was brought about by the induction of AOS as a key enzyme rather than by stimulation of its downstream partner AOC.

The earlier activation of jasmonate synthesis in *A. hypochondriacus* × *hybridus* is mirrored by an earlier activation of jasmonate responses as indicated by an earlier induction of transcripts for jasmonate ZIM-domain 1 (JAZ1). The respective gene product acts as a receptor for the active jasmonate conjugate with isoleucine and, upon binding, is proteolytically degraded, resulting in a de-repression of jasmonate-responsive genes including JAZ genes themselves. Thus, the induction of JAZ transcripts can be used as a proxy for the formation of bioactive jasmonate–isoleucine conjugates [30]. For *kiwicha*, three of the four tested genotypes accumulated JAZ1 only during full maturation (Figure 5D), i.e., at a time point when, due to autumn, temperatures had already dropped conspicuously. For genotype 2496 (*A. caudatus* cv. Marron), there was a transient induction during stage 2 (correlating with a transient accumulation of transcripts for Lipoxygenase (Figure 5A) and AOS (Figure 5B)) at the same developmental state.

Overall, the later activation of jasmonate biosynthesis and signalling in *A. caudatus* correlates with a significantly higher level of linoleic acid (compare Figure 5A–D with Figure 4B). In contrast, in *A. hypochondriacus* × *hybridus*, the jasmonate pathway is activated already at fairly warm temperatures accompanied by reduced steady-state levels of linoleic acid.

We wondered how these differences in the partitioning of precursors between the synthesis of unsaturated fatty acids versus oxylipins were reflected in the response of genes for cold signalling and adaptation. The perception of low temperature deploys a signal transduction that targets ICE1, a master switch that is continuously produced and degraded in the proteasome with a high turnover [33]. Under cold stress, the proteolytic degradation of ICE1 becomes inhibited, such that the steady-state levels of the ICE1 protein will increase and activate the CBFs, such that COR transcripts will be induced. We, therefore, selected the master switch ICE1, the cold acclimation regulator CBF4, and the cold-adaptive dehydrin Xero2 to monitor the cold response of the five genotypes (Figure 5E–G).

We observed that ICE1 transcripts accumulated to higher levels in *A. caudatus* as compared to *A. hypochondriacus* × *hybridus*, albeit at different time points and with different amplitudes. The most salient difference was seen in the *A. caudatus* genotype Peter 2 (8301), where ICE1 transcripts were strongly elevated already during stage 1 and were strongly induced during stage 3. Additionally, in the *A. caudatus* genotype Oscar Blanco (9445), this transcript was abundant from stage 2, whereas the induction was weak (8300) or late (2496) in the other *A. caudatus* genotypes. It should be kept in mind that ICE1 is regulated on the post-translational level but needs to be continuously replenished due to its strong turnover, such that there must be feedback from cold signalling upon the transcription of this gene. Elevated levels of ICE1 transcripts might, thus, report a more vigorous cold response in *A. caudatus*, which, as pointed out earlier, flowered significantly later and, thus, experienced lower temperatures during seed development as compared to *A. hypochondriacus* × *hybridus* (Appendix A).

However, the strong and early induction of JAZ1 expression has a more immediate impact on the ICE1 protein itself, since in Arabidopsis thaliana, JAZ1 has been shown to repress the binding of ICE1 to target promoters, including the CBF genes themselves [34]. This would imply that the accumulation of CBF genes should be downmodulated in *A. hypochondriacus* × *hybridus* as compared to *A. caudatus*.

To test this implication, we addressed, therefore, the behaviour of Cold Box Factor 4 (CBF4), which in various plants has been shown to activate cold acclimation as well as the dehydrin Xero2, a COR protein linked with freezing tolerance. Transcripts for CBF4 accumulate vigorously in all *A. caudatus* accessions but 2496 from stage 2, but only weakly in *A. hypochondriacus* × *hybridus* (Figure 5F). One might argue that this difference might be linked with the earlier flowering in *A. hypochondriacus* × *hybridus*, which is, therefore, facing lower levels of cold stress. This hypothesis is falsified by the fact that transcripts for lipoxygenase, reporting general oxylipin synthesis, and also transcripts specific for jasmonate synthesis and signalling are all elevated in *A. hypochondriacus* × *hybridus* already from stage 1 (Figure 5A–D), indicating that this genotype, already at milder temperatures, experiences low-temperature stress. However, it fails to deploy adaptive gene expression. This conclusion is confirmed by the pattern observed for the downstream COR transcripts for Xero2 (Figure 5G). These accumulate very vigorously in stage 3 of all *A. caudatus* genotypes except 2496. Although induction can be seen in *A. hypochondriacus* × *hybridus* as well, it occurs at a much lower amplitude. Among the Cold Box Factors, CBF4 differs by a slower but more persistent induction by cold stress (grapevine [87,88]; strawberry [89]; *Arabidopsis thaliana* [90]). Upon overexpression, it confers freezing tolerance (grapevine [91]; tobacco [92]; cotton [93]) and, thus, is thought to be a central activator of cold adaptation. Although other CBFs are always localised in the nucleus, CBF4 is found in the cytoplasm but is imported in response to cold stress [92]. The COR protein Xero2 (also known as Lti30) belongs to the acidic subclass of dehydrins, proteins that are induced in many plants in response to cold stress and are thought to act as cryoprotectants [94]. By virtue of a lysine-rich repeat sequence forming an amphipathic α-helix, it can electrostatically interact with the negatively charged phosphate groups of membrane lipids and, thus, stabilise the lamellar structure of membranes challenged by water loss under drought but also under freezing [95,96].

Summarising, the response patterns of genes for oxylipin biosynthesis, jasmonate signalling, cold signalling, and cold adaptation show clearly that *A. hypochondriacus* × *hybridus* experiences cold stress while failing to deploy cold adaptation. In contrast, *A. caudatus* can efficiently activate cold adaptation, although it seems more mildly affected by cold stress. This genetic difference is even more remarkable when one takes into account that *A. caudatus* flowers later in the year and consequently is exposed to lower temperatures.

## 3. Materials and Methods

### 3.1. Plant Material

A total of 5 genotypes of Amaranthus species were used in this study (Table 1). Of these genotypes, 4 were classified as *A. caudatus* (Peter 1, Peter 2, Oscar Blanco 2, Marron), and 1 as *A. hypochondriacus* × *hybridus* (K 432). The amaranth genotype K 432 (ID 8039) was developed in the United States by crossing *A. hypochondriacus* and *A. hybridus* [97]. Oscar Blanco 2 (ID 9445) is an accession cultivated in the Mollepata province and belongs to the economically important landrace Oscar Blanco, which is widespread in Peru due to its light colour and favourable popping characteristic that is the preferred form of consumption [98]. Marron (ID 2496) is a genotype from the amaranth germplasm bank in Universidad Nacional de San Antonio Abad del Cusco (UNSAAC), which, due to its higher oil content, has been distributed among farmers during recent years to support its commercialisation. The genotypes Peter 1 (ID 8300) and Peter 2 (ID 8301) are native caudatus landraces bought in a farmer’s market in Cusco city in 2013 from an indigenous woman, who could provide amaranth grains from two locations in the Cusco region: Peter 1 originated from Valle Sagrado (sacred valley) in Urubamba province, whereas Peter 2 was from the district San Salvador in the province Calca.

### 3.2. Cultivation of the Plant Material

The five genotypes were evaluated with respect to their agro-phenological traits in a field experiment under temperate environmental conditions in the Botanical Garden of the Karlsruhe Institute of Technology (KIT) in the vegetation season of 2020. Prior to transfer to the field plot, seeds were pre-planted in 100-well trays in the greenhouse on 15 April 2020. Six surface-sterilised seeds were sown per well and then thinned in the greenhouse to one plant per well after germination. A total of 50 wells were planted per genotype, resulting in 50 seedlings. The wells were filled with a mixture of a 1:1:1 peat moss/perlite/soil mixture and raised in the greenhouse at 25 ± 3 °C with a 12 h photoperiod. Three weeks after germination, the seedlings were transplanted to the field plot. To ensure a good plant stand, the distance between individual amaranth seedlings within the row was 40 cm. The length of each plot was 2.5 m, and the distance between the rows was 0.65 m. The seedlings were transplanted in three randomised experimental blocks, with each block containing three rows for each genotype. The two outer rows served as buffers, whereas all measured parameters were collected from the plants in the middle row. Based on soil analysis, the recommended amounts of fertiliser were supplemented as 100 g/m^2^ organic fertilizer (Hauert Hornoska^®^ Special, Zossen, Germany) and 90:60:40 kg/ha of NPK. Soil humidity was kept to 70% to 80% of field capacity by irrigation throughout the period from transplanting to seed maturity. Karlsruhe is located in the Rhine Valley in the southwest of Germany (latitude: 49°0′24.8004″ N, longitude: 8°24′13.1508″ E), with an average elevation of 119 m above sea level (based on the World Geodetic System 1984 datum). Temperatures range from around −1 °C during winter to 26 °C during summer in a temperate oceanic climate. Temperature and rainfall were monitored for the duration of the experiment, and monthly average values are presented in Appendix A.

### 3.3. Phenotyping and Morphological Evaluations

For morphological observations, three plants located in the centre of a second row in each plot were sampled randomly to record flowering time (day), plant height (cm), length of inflorescence (cm), grain yield (g/plant), 1000-seed weight (mg), and cross area of seeds (mm^2^). The length of inflorescence was measured from the lower flower to the tip of the highest flower in each plant. Seed cross area was determined from three randomly selected seeds for each of the three individuals per replicate, recording the seed using a Keyence VHX-950F digital microscope (Keyence, Neu-Isenburg, Germany), and measuring the cross section using the area tool of ImageJ (https://imagej.nih.gov/ij/). To ensure comparability, the seeds were oriented such that the image plane was parallel to their flatter axis. Qualitative traits such as flower colour and seed colour were documented at the flowering and maturity stages by recording seeds and flowers of each genotype using a Keyence VHX-950F digital microscope. Seeds were harvested manually by pruning shears when all seeds had reached full maturity. The harvested panicles were dried in a hoop house for two weeks and threshed manually. Threshed seeds were winnowed and evaluated for grain yield, 1000-seed weight, and seed colour.

### 3.4. Oil Extraction

Amaranth oil was extracted from mature seeds according to [67] with minor modifications. Around 1–2 g of seeds was homogenised with a Waring blender. In brief, 0.5 g of homogenised seed material was transferred into 2 mL reaction tubes and filled up with the extraction agent hexane/acetone (v:v 75:25). The samples were then sonicated at 100% amplitude for 1 min, shaken for 1 h at about 25 °C, and spun down at 25,000 rpm at 25 °C for 3 min, and then the supernatant was transferred into volumetric flasks. The residue was re-extracted in the same way, and the two extracts were pooled to obtain a larger amount of extracted oil. The extracts were dried with sodium sulfate, collected with a disposable syringe, and filtered through a CHROMAFIL^®^ PET-20/15 MS filter (pore size 0.25 μm). The solvent was evaporated on a rotary evaporator at a water bath temperature of 40 °C and an initial pressure of 300–400 mbar to remove the bulk of the solvent. Subsequently, the pressure was gradually reduced to 50 mbar until the solvent had been fully removed. The extracted oil was stored in 1 mL flasks at 4 °C for up to 5 days until analysis. To suppress oxidation, butylated hydroxytoluene (200 ppm) was added.

### 3.5. Analysis of Fatty Acids by GC-FID

In order to determine the fatty acid profile of the extracted oil, a trans-esterification for converting fatty acids to fatty acid methyl esters (FAME) was carried out using the methylation reagent trimethylsulfonium hydroxide (TMSH). For this, between 0.02 g and 0.03 g of the extracted oil was weighed into a flask and filled up to 1 mL with hexane. Aliquots of 100 µL of this solution were pipetted into a GC vial, adding 50 µL of TMSH and subsequently 50 µL of ethyl acetate. The sample was then mixed and allowed to equilibrate for 30 min before proceeding with the GC analysis. A total of 1μL of each sample solution was injected into the GC System, an Agilent 7890B equipped with a DB-FATWAX UI (Agilent Technologies Inc. G3903-63008) column (30 m × 0.25 mm × 0.25 μm). Fatty acids were eluted under a programme with a starting temperature of 100 °C, heating at 15 °C min^−1^ to 200 °C (held for 50 min), followed by a second heating step at 50 °C min^−1^ to 240 °C (held for 10 min), a flow rate of 1.2 mL/min (32 cm/s), injector temperature of 240 °C, and FID-detector temperature set to 245 °C. The extraction was conducted from two biological replicates, each in technical duplicates.

### 3.6. DNA Extraction for the Whole Genome Sequencing

The genomic DNA of two genotypes, *A. hypocondriacus* × *hybridus* cv. K 432 and *A. caudatus* cv. Peter 1 was extracted using cetyl trimethyl ammonium bromide (CTAB) according to [99] from 100 mg of frozen leaf tissue, ground by a TissueLyzer. In brief, the powder was treated with 900 µL of boiled extraction buffer (1.5% *w*/*v* CTAB) containing 10 µL/mL β-mercaptoethanol and incubated for one hour at 65 °C. The samples were mixed with 630 µL of chloroform/isoamylalcohol (24:1), shaken horizontally for 15 min, and subsequently centrifuged for ten minutes (17,000× *g*). The upper aqueous phase (which contains the DNA) was transferred into a fresh 2 mL reaction tube, and the DNA was precipitated with 2/3 *v*/*v* of ice-cold isopropanol. The purified DNA was collected by centrifugation (10 min, 17,000× *g*). The pellet was washed with 1 mL 70% EtOH after removing the EtOH dried in a vacuum centrifuge for 15 min and finally dissolved in 50 µL nuclease-free H_2_O (containing 5 µg RNAse A). The concentration and purity of the eluted DNA were determined spectrophotometrically using a NanoDrop (ND-100, peqlab).

### 3.7. DNA Sequencing, Assembly, Annotation

The genotypes *A. caudatus* cv. Peter1 (8300) and *A. hypochondriacus* × *hybridus* cv. K432 (8039) were subjected to 150 bp paired-end Illumina whole-genome sequencing on a NovaSeq platform. The files for accession 8300 comprised 99,496,035 read pairs, and those for accession 8039 comprised 101,129,180 read pairs. As judged from the FastQC reports, the raw reads were of very good quality and were void of discrepancies and adapter contaminations. To map the genome, we made use of the reference genome for *A. hypochondriacus* PI 558,499 cv. “Plainsman” [78] version v2.0 with a genome size of 410 Mbp accessed through Phytozome v12 (Phytozome: phytozome.jgi.doe.gov/pz/portal.html#!info?alias=Org_Ahypochondriacus_er, accessed on 6 June 2021). The raw reads were mapped on this genome using BWA v0.7.17 [100]. More than 97% reads of accession 8300 (Peter1) and 98% reads of accession 8039 (K 432) were mapped onto the reference genome. The aligned BAM files were further checked for duplicate reads (around 22% in both samples) which were located and removed using Picard v2.23.4 (https://broadinstitute.github.io/picard/, accessed on 3 March 2020) prior to further downstream analysis (Appendix A). Quality check reports of these BAM files were generated using QualiMap v2.2.1 (http://qualimap.conesalab.org/, accessed on 9 September 2022), and these finalised BAM files were then used for variant calling and further downstream analysis. To understand the differences between these varieties in comparison to the reference, variant calling was performed using the ‘HaplotypeCaller’ v4.1.2 of the GATK software (https://gatk.broadinstitute.org/hc/en-us, accessed on 2 February 2020). The raw variant (VCF) files were filtered using the filters Quality-by-depth (QD < 5.0) or mapping quality (MQ < 50.0), and low-quality variants were removed using the ‘VariantFiltration’ routine of the GATK software (https://gatk.broadinstitute.org/hc/en-us, accessed on 2 February 2020).

After the two genomes had been analysed for their overall differences, individual single nucleotide polymorphisms (SNPs) were investigated using SnpEff v5 (https://pcingola.github.io/SnpEff/, accessed on 2 February 2020). Subsequently, pseudogenes were generated for all genes, including a prioritised subset of genes that were of interest based on the reference sequence with variants replaced by respective genotypes as well as using GATK v4.1.2 (https://gatk.broadinstitute.org/hc/en-us, accessed on 2 February 2020).

### 3.8. RNA Extraction, cDNA Synthesis and Quantitative Real-Time PCR

To study the developmental dynamics of metabolic genes involved in fatty acid biosynthesis [28], jasmonate biosynthesis and signalling [30], as well as genes linked with cold-stress signalling and response [101], flower samples were collected from outdoor-grown plants at three specific developmental stages, namely, at anthesis (S1), milky seed stage (S2), and seed maturity (S3), immediately frozen in liquid nitrogen, and stored at −80 °C until further analysis. The samples were collected from the middle part of the inflorescence. Total RNA was isolated using the Spectrum™ Plant Total RNA Kit (Sigma, Schnelldorf, Germany) according to the instructions of the manufacturer from a small amount of tissue ground to a powder (Tissue Lyzer, Qiagen, Hilden, Germany). The extracted RNA was reversely transcribed into cDNA by M-MuLV Reverse Transcriptase (New England Biolabs, Frankfurt am Main, Germany) using 1 μg of total RNA as a template. Real-time (qPCR) was performed with the CFX96 Touch™ Real-Time PCR Detection System from Bio-Rad Laboratories GmbH (Munich) using an SYBR Green dye protocol according to [102]. Transcript levels between the different samples were compared using the ΔCt method [103] and normalised to actin (*Ah.Act*) as a housekeeping gene. Each data point represents three biological replicates, each conducted in technical triplicates. The functional context of the probed genes is shown in Figure 6; the accession numbers of these genes and the sequences of the oligonucleotide primers are given in Appendix A.

### 3.9. Statistical Analysis

Phenotypic data were subjected to individual ANOVA for different characters in order to assess the variability among the genotypes and standard error of the treatment means using PROC ANOVA in the SAS program (SAS Institute, Inc., Cary, NC, USA). The least significant difference (LSD) test was used to carry out post hoc comparisons of differences among means, applying a significance threshold of *p* < 0.05 (PROC MEANS).

## 4. Conclusions

In conclusion, our research focuses on the morphological, biochemical, and molecular differences between five amaranth genotypes belonging to the *A. caudatus* and *A. hypocondriacus* species under temperate environmental conditions. The *A. caudatus* genotypes were late in flowering, exposing them to cold stress during the milky and mature seed developmental stages, resulting in a significant reduction in grain yield per plant when compared to *A. hypochondriacus* × *hybridus* genotype 8039. The *A. hypochondriacus* × *hybridus* genotype 8039 had higher saturated fatty acid compositions in seed oil, whereas the late-flowering *A. caudatus* had higher polyunsaturated fatty acid compositions, such as linoleic and α-linolenic acids, which contribute to maintaining the fluidity of biological membranes in plants under low temperatures as a biochemical mechanism to adapt to cold climate at flowering times in the high Peruvian region compared to the Mexican region. The differences in cold adaptation and flowering times between the genotypes studied had a significant impact on the transcriptomic level of all genes involved in the ω3 and Jasmonic acid biosynthetic pathways, as well as cold tolerant-related genes, at the initiative, milky, and maturity seed developmental stages. The fatty acid desaturase and elongase genes were substantially expressed in the late-flowering *A. caudatus* genotypes due to a decline in temperature during the seed developing period. For *A. hypochondriacus* × *hybridus* (8039), there was a greater shift toward the jasmonic acid biosynthetic pathway than the ω3 biosynthetic pathway. However, when compared to late-flowering *A. caudatus* genotypes, which are more adapting to cold climates at flowering time in the higher Peruvian area, the transcript quantity was decreased in subsequent steps of the JA pathway as well as in cold-tolerant-related genes of *A. hypochondriacus* × *hybridus* genotype (8039) at the maturity seed developmental stage. A thorough investigation of the transcriptomic profiling of the genes involved in the ω3 pathway and cold tolerance is essential to improving oil quality and stress resistance in plants. More research is needed to identify and analyse the gene expression of Δ5 desaturase, Δ5 elongase, and Δ4 desaturase. This will serve as the basis for modifying the fatty acid compositions of membranes in order to improve the vitality and vigour of oilseed crops.

## Figures and Tables

**Figure 1 ijms-24-06215-f001:**
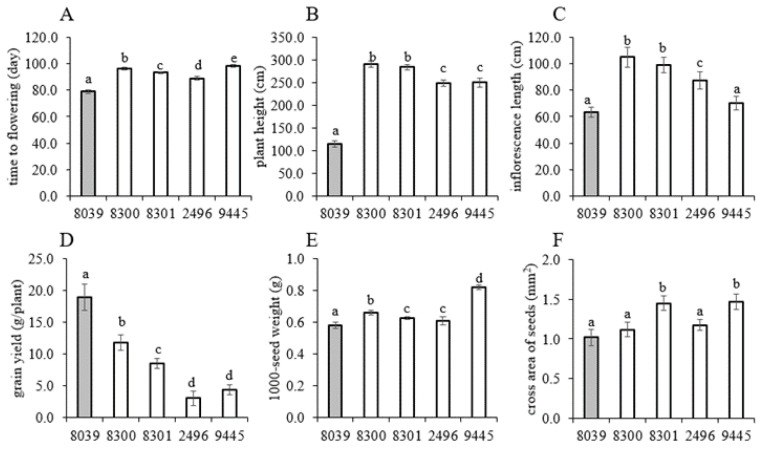
Mean values of some agro-morphological traits recorded from 5 amaranth genotypes evaluated under temperate environmental conditions in Southwest Germany during the season of 2020. (**A**) Days to flowering, (**B**) plant height (cm), (**C**) inflorescence length (cm), (**D**) grain yield (g/plant), (**E**) 1000-seed weight, and (**F**) cross area of seeds (mm^2^). Data represent the mean of three biological replicates with error bars representing the standard error. Briefly, 8039 = *A. hypocondriacus* × *hybridus* cv. K 432; 8300 = *A. caudatus* cv. Peter 1; 8301 = *A. caudatus* cv. Peter 2; 2496= *A. caudatus* cv. Marron; and 9445= *A. caudatus* cv. Oscar Blanco 2. The white bars indicate the *A. caudatus* accessions. Bars with different letters are significantly different at *p* ≤ 0.05 based on ANOVA, and least significant difference (LSD) test.

**Figure 2 ijms-24-06215-f002:**
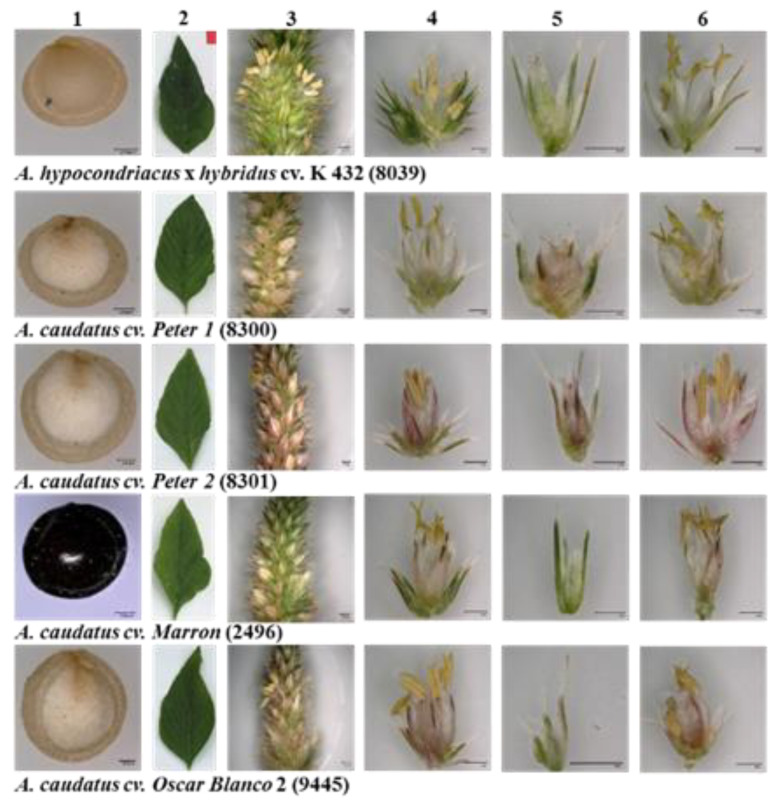
Representative images of (**1**) seed, (**2**) leaf, (**3**) panicle, (**4**) glomerulus, (**5**) pistillate flower, and (**6**) staminate flower of five amaranth genotypes evaluated under temperate environmental conditions in Southwest Germany during the season of 2020. Seeds and flower details were recorded with a magnification of 150×.

**Figure 3 ijms-24-06215-f003:**
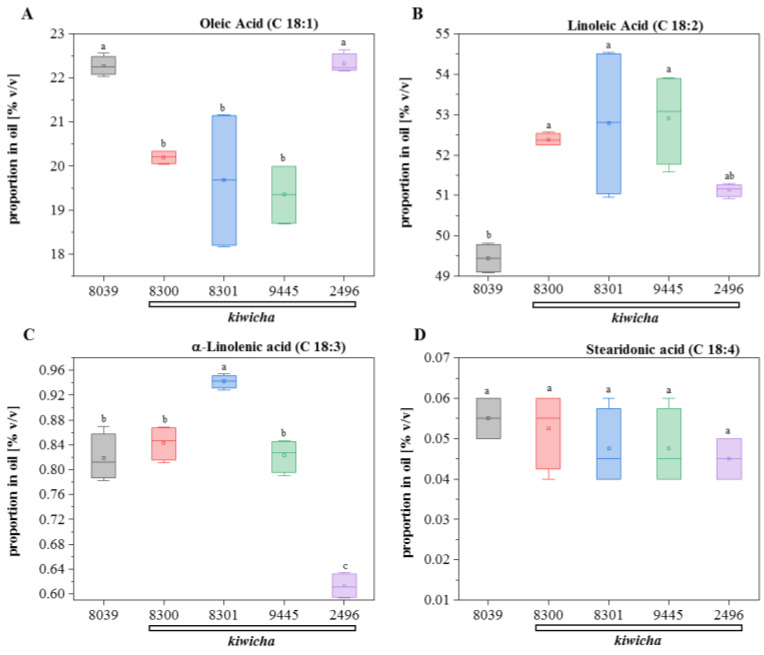
Relative abundance of unsaturated fatty acids (as % *v*/*v*) in oil extracted from mature seeds of the 5 amaranth genotypes. (**A**) Oleic Acid, (**B**) linoleic Acid, (**C**) α-linolenic Acid, and (**D**) stearidonic Acid. Briefly, 8039 = *A. hypocondriacus* × *hybridus* cv. K 432 (grey); 8300 = *A. caudatus* cv. Peter 1 (red); 8301 = *A. caudatus* cv. Peter 2 (blue); 2496 = *A. caudatus* cv. Marron (green); and 9445 = *A. caudatus* cv. Oscar Blanco 2 (purple). Data represent median and quartiles for four individual measurements (two individual plants that were each extracted twice). Genotypes with different letters are significantly different (ANOVA, least significant difference (LSD) test, *p* ≤ 0.05).

**Figure 4 ijms-24-06215-f004:**
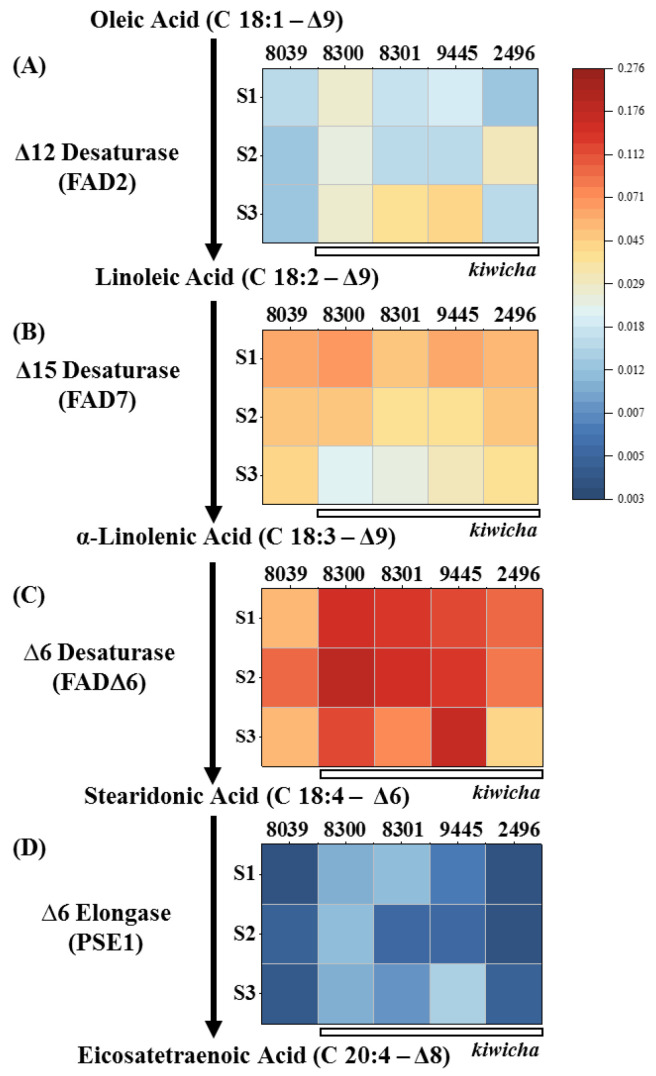
Steady-state transcript levels for genes involved in ω3 fatty acid desaturase biosynthesis in the 5 amaranth genotypes over three developmental stages of the seed. (**A**) Δ12 desaturase, (**B**) Δ15 desaturase, (**C**) ∆6 desaturase; (**D**) ∆6 elongase. The plants were raised under temperate environmental conditions in southwestern Germany during the season of 2020. S1 initiating seed development, S2 milky stage, and S3 seed maturity. Briefly, 8039 = *A. hypocondriacus* × *hybridus* cv. K 432; 8300 = *A. caudatus* cv. Peter 1; 8301 = *A. caudatus* cv. Peter 2; 2496 = *A. caudatus* cv. Marron; and 9445 = *A. caudatus* cv. Oscar Blanco 2. Transcript levels were normalised against actin as internal standard. Values are means of three biological replicates each in technical triplicate. The colour scale gives the relative expression based on DC_t_ values.

**Figure 5 ijms-24-06215-f005:**
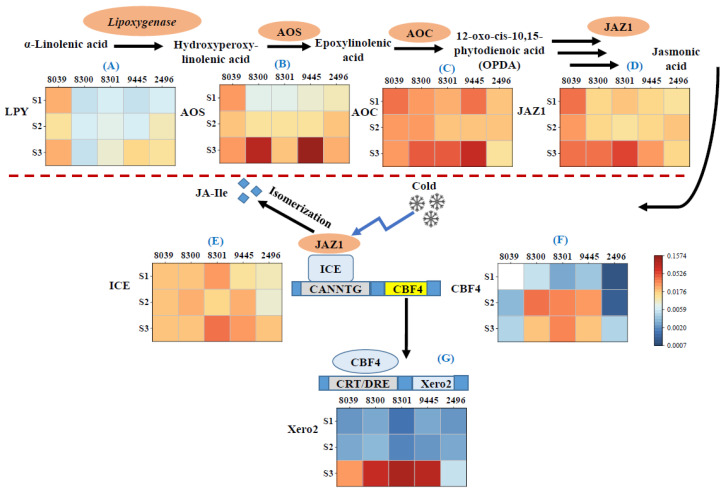
Steady-state transcript levels for genes involved in jasmonate biosynthesis and cold signalling/response in the 5 amaranth genotypes over three developmental stages of the seed. (**A**) Lipoxygenase; (**B**) allene oxide synthase; (**C**) allene oxide cyclase; (**D**) jasmonate ZIM domain protein 1; (**E**) inducer of CBF expression 1; (**F**) C-repeat-binding Factor 4; (**G**) dehydrin Xero2/low-temperature-induced protein. The plants were raised under temperate environmental conditions in southwestern Germany during the season of 2020. S1 initiating seed development, S2 milky stage, and S3 seed maturity. Briefly, 8039 = *A. hypocondriacus* × *hybridus* cv. K 432; 8300 = *A. caudatus* cv. Peter 1; 8301 = *A. caudatus* cv. Peter 2; 2496 = *A. caudatus* cv. Marron; and 9445 = *A. caudatus* cv. Oscar Blanco 2. Transcript levels were normalised against actin as internal standard. Values are means of three biological replicates each in technical triplicate. The colour scale gives the relative expression based on DC_t_ values.

**Figure 6 ijms-24-06215-f006:**
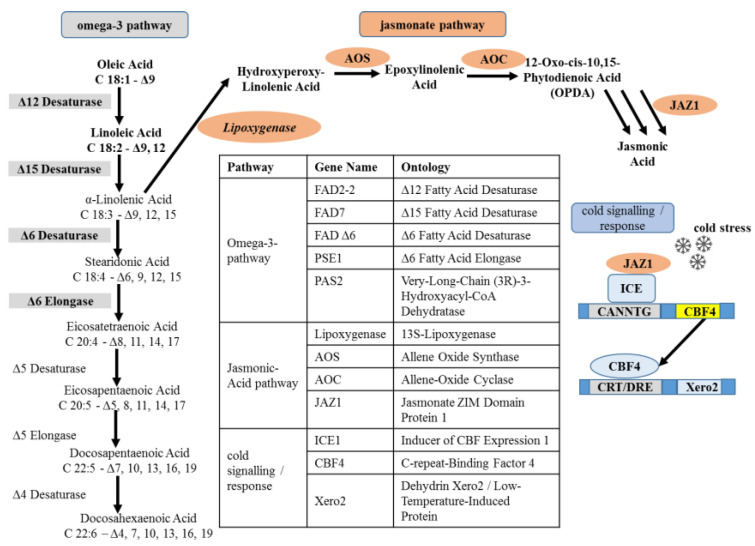
Metabolites and genes of the ω3 fatty-acid pathway according to [28], along with the concurrent pathway for jasmonic acid synthesis pathway according to [104] and cold stress markers according to [101]. Genes investigated during the current study are highlighted and shown along with their functions in the table. The respective oligonucleotide primer sequences for these genes are given in Appendix A.

**Table 1 ijms-24-06215-t001:** Origin of the 5 amaranth genotypes included in this study.

ID KIT	Genotype’s Names	Origin	*Provider*
9445	*A. caudatus* cv. Oscar Blanco 2	Mollepata province, Cusco region, Peru	CME, Peru
2496	*A. caudatus* cv. Marron	Cusco province, Cusco region, Peru	UNSAAC, Peru
8300	*A. caudatus* cv. Peter 1	Urubamba province, Cusco region, Peru	BGU, KIT, Germany
8301	*A. caudatus* cv. Peter 2	Calca province, Cusco region, Peru	BGU, KIT, Germany
8039	*A. hypocondriacus* × *hybridus* cv. K 432	USA	BGU, HU, Germany

UNSAAC = Universidad Nacional de San Antonio Abad del Cusco, CME = Cusco Mara Eirl, BGU = Botanical Garden of the University, KIT = Karlsruhe Institute of Technology, HU = Hohenheim University.

## Data Availability

Data can be made available upon reasonable request.

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
