# Peer review of "Peruvian Amaranth (kiwicha) Accumulates Higher Levels of the Unsaturated Linoleic Acid"

_ijms, 2023, doi:10.3390/ijms24076215_

Round 1

Reviewer 1 Report

In the present manuscript, the authors compared the agro-morphological traits of different amaranth genotypes. They also compared the fatty acid composition of different varieties along with molecular interpretations. The following suggestions should be incorporated for improvement of manuscript.

- It is suggested to modify the title. It should be self explanatory for the content of manuscript.

- Ratioale of study is not clear in the introduction. It should be elaborated with clear hypothesis. So may concepts are discussed in introduction section, which could confuse the readers so it is suggested to clearly elaborate the hypothesis.

- In table s1: Significance of results compared to which group??? Also legend suggested that values followed by same letter are not significantly different however variety 9445 and 8300 does not seems to be significantly different.

- Also in the result section, the results are not properly discussed. It is suggested to justify the results with proper citations. 

-In fig S2, first variety is 3089 but it is not in the text, why???

- Also what is the significance of table s2. Authors claim, highly significant values, but significance a s compare to what??? The data is very cofusing.

- In fig 1, it is suggested to indicate the significance on individual bar better understanding of readers.

- Plat height, inflorescence length, 100 seed weight and cross sectional area of seeds is less for A hypocondraicus variety but grain yield is more for this, Justifications of results should be there. 

-Table s4 is again confusing, what does it indicates?? The fatty acid composition in oe variety or 5 variety?? The value given is average one or significance level?? If significance level then as compared to what?? Leged of table is very confusing.

Overall, manuscript text is very confusing. Results are not properly justified. Different varieties are termed differently at various places, no uniformity. 

The authors should include the suggestions for improvement of mauscript. 

Author Response

Reviewer 1

Comments and Suggestions for Authors

In the present manuscript, the authors compared the agro-morphological traits of different amaranth genotypes. They also compared the fatty acid composition of different varieties along with molecular interpretations. The following suggestions should be incorporated for improvement of manuscript.

- It is suggested to modify the title. It should be self explanatory for the content of manuscript.

Response:  We have changed the title to “Peruvian Amaranth (kiwicha) accumulates higher levels of the unsaturated linoleic acid”

- Ratioale of study is not clear in the introduction. It should be elaborated with clear hypothesis. So may concepts are discussed in introduction section, which could confuse the readers so it is suggested to clearly elaborate the hypothesis.

Response:  We see from the response that our scope section was too implicit. We have now rewritten the last paragraph to explicitly describe our working hypothesis that Peruvian Amaranth, due to its domestication in high altitude, is endowed with a higher cold tolerance linked with accumulation of higher levels of unsaturated fatty acids and a lower level of oxylipins.

- In table s1: Significance of results compared to which group??? Also legend suggested that values followed by same letter are not significantly different however variety 9445 and 8300 does not seems to be significantly different.

Response:  We have used the least significant difference (LSD) test in the context of the analysis of variance (ANOVA), when the difference between the varieties means is significant. The basic idea of the test, as you are aware, is to compare the varieties taken in pairs after the null hypothesis has already been rejected (Table S2). With respect to the second part of your question, LSD test is the most “sensitive” Post Hoc Test and most unlikely to miss significant mean group difference. However, we have re-analysed the data again to confirm the result and we found that the two varieties are significantly different, and we have explained more clearly, what was compared to what in Table S1.

- Also in the result section, the results are not properly discussed. It is suggested to justify the results with proper citations. 

Response:  It is not entirely clear, what the reviewer wants us to do. Please keep in mind that this work is the first study addressing the biosynthesis of unsaturated fatty acids, jasmonate metabolism, and cold signalling in Amaranth. There is not much to be quoted to justify our result. Moreover, a novel study should be justified by data in the first place, not by recapitulating, what others have been doing. However, in order to address this point, we have added additional references on fatty-acid metabolism under cold stress and jasmonate biosynthesis, albeit from other species.

-In fig S2, first variety is 3089 but it is not in the text, why???

Response:  Thank you so much for catching this error, which we have now corrected.

- Also what is the significance of table s2. Authors claim, highly significant values, but significance as compare to what??? The data is very confusing.

Response:  Thank you for yours commend. Before doing the LSD test, we have done F-test (Table S2) to see if the null hypothesis H0 is rejected or not. The null hypothesis was rejected (the mean values of a single trait for the five genotypes are NOT equal) because the highly significant difference between the five genotype means. We are not comparing here (in ANOVA); we are just testing our model. In other way, we found that the mean values of the five genotypes for all the traits are highly significant, which allow us to go for multiple comparison tests such as LSD test. We have already done that in our result. Since we see that our description was too laconic and implicit, we have now spelled out in more detail, what Table S2 is actually showing.

- In fig 1, it is suggested to indicate the significance on individual bar better understanding of readers.

Response:  While we understand the intention, the LSD test is not comparing pairwise, but compares the whole set of samples simultaneously. The alternative would be to test differences pairwise individually, but this would make the figure extremely crowded and hard to read. To address the request for more transparency, we have spelled out in the figure legend more explicitly, what the letters mean.

- Plat height, inflorescence length, 100 seed weight and cross sectional area of seeds is less for A. hypocondraicus variety but grain yield is more for this, Justifications of results should be there. 

Response:  The reason for this is that A. hypochondriacus is flowering earlier, while A. caudatus, under the conditions in Southwest Germany is flowering later, such that it becomes already limited by falling autumn temperatures. While we have discussed this point already in the text, it seems that it was not explicit enough. We have, therefore, inserted now a short statement, where we explicitly mention this point raised by the reviewer and give the reason for this difference.

-Table s4 is again confusing, what does it indicates?? The fatty acid composition in one variety or 5 variety?? The value given is average one or significance level?? If significance level then as compared to what?? Leged of table is very confusing.

Response: We regret that the statistical treatment of the data by analysis of variance (ANOVA) makes it difficult to read the table, but there is no alternative to this complexity. The rules of statistics are as they are. To help the reviewer to understand this table, we give the following explanation: The mean values in the table (the fourth row) are the values of overall mean for the five varieties. The values in the genotype (the first row) are the mean square value (MS) which was calculated from the sum of square values. This is the standard approach of showing the ANOVA results. The stars on the MS values give the significance level based on F-test. Our null hypothesis is that there is no difference among the five varieties means. If any variety differs significantly from the overall varieties mean, then the ANOVA will report a statistically significant result. To know which variety differs significantly from the overall mean, we have done Post-ANOVA test (LSD test in our data). But we see that for readers that are not familiar with statistics, we need to explain Table S4 in more detail, what we have done now.

Overall, manuscript text is very confusing. Results are not properly justified. Different varieties are termed differently at various places, no uniformity. The authors should include the suggestions for improvement of mauscript. 

Response: While we do not understand, what the reviewer refers to, because this statement is not specific, we have now reworked the text and have used the same designations for the genotypes throughout in a way that we give the taxon name and the ID.

Reviewer 2 Report

This is a well designed study and a well written MS.

I would suggest some edits to the authors:

1. please avoid using the term "super food"; it's not substantiated.

2. an explanation from a plant biochemical point of view should be given on the reasons why this plant desaturates fatty acids, i.e., does this lead to better defence - via the jasmonate pathway? (also, does it lead to other phenomena? how this desaturation affects the overall nutritional value of the plant?)

3. some discussion on the nutritional role of highly unsaturated fatty acids should be added. This will render the MS even stronger. Some comparisons could be provided to the levels of unsaturated fatty acids in other plants that serve as staple diet. Also, what is the anti-inflammatory role of these unsaturated fatty acids? Some references should be added on this point, suggested reference: https://www.mdpi.com/2304-8158/11/16/2442

The authors would also need to edit the role of omega3/6 in relation to CVD! Today, we know that omega3s do not lower cardiovascular risk - https://pubmed.ncbi.nlm.nih.gov/35275889/ 

Inflammation is the underlying cause of CVD and other chronic diseases, suggested references: 

Nutrients | Free Full-Text | Cholesterol versus Inflammation as Cause of Chronic Diseases (mdpi.com) 

and

https://www.mdpi.com/2072-6643/10/5/604

Happy to review the revised MS. 

Author Response

Reviewer 2

This is a well designed study and a well written MS.

I would suggest some edits to the authors:

  1. please avoid using the term "super food"; it's not substantiated.

Response: The term “super food” has been deleted from the MS, and instead Amaranth nutritional value has been highlighted.

  1. an explanation from a plant biochemical point of view should be given on the reasons why this plant desaturates fatty acids, i.e., does this lead to better defence - via the jasmonate pathway? (also, does it lead to other phenomena? how this desaturation affects the overall nutritional value of the plant?)

Response: We see from this comment that we need to be more explicit. The main point is that the desaturation pathway and the oxylipin pathway compete for the same precursor (alpha-linolenic acid). Thus, the plant has to render a decision. Desaturation of fatty acids is an adaptation to cold because it helps to sustain membrane fluidity. The use of unsaturated fatty acids to generate jasmonates as stress signals will, thus, impair membrane adjustment, and can, therefore, be seen as indicator that the plant is under stress. The difference between kiwicha is that it goes for lipid desaturation because it is adapted to this, while the Mexican Amaranth is not and responds with producing a stress signal. We have now integrated this point into our working hypothesis (see also our response to reviewer 1) and once more, as explicit as here, into the conclusions.

  1. some discussion on the nutritional role of highly unsaturated fatty acids should be added. This will render the MS even stronger. Some comparisons could be provided to the levels of unsaturated fatty acids in other plants that serve as staple diet. Also, what is the anti-inflammatory role of these unsaturated fatty acids? Some references should be added on this point, suggested reference: https://www.mdpi.com/2304-8158/11/16/2442

The authors would also need to edit the role of omega3/6 in relation to CVD! Today, we know that omega3s do not lower cardiovascular risk - https://pubmed.ncbi.nlm.nih.gov/35275889/ 

Inflammation is the underlying cause of CVD and other chronic diseases, suggested references: 

Nutrients | Free Full-Text | Cholesterol versus Inflammation as Cause of Chronic Diseases (mdpi.com) 

And https://www.mdpi.com/2072-6643/10/5/604

Response: While the nutritional details and medicinal activity of PUFAs are not the scope of this study, we have now related our data of the linolenic-acid contents in kiwicha with contents reported for other staple crops. We have downtoned our statements on the link with Cardiovascular Diseases and have quoted the suggested reference, although it is just a meta-study and not overly conclusive, since it does not address any mechanisms. Also, we have included evidence for the anti-inflammatory role of Omega-3 fatty acids and a link to mitigation of depression coming from the recent study by Giacobbe et al. 2020.

Round 2

Reviewer 1 Report

The manuscript has been revised as per the suggestions and can be accepted. 

Reviewer 2 Report

the MS can now be accepted